

# Sherlock: Towards Multi-scene Video Abnormal Event Extraction and Localization via a Global-local Spatial-sensitive LLM

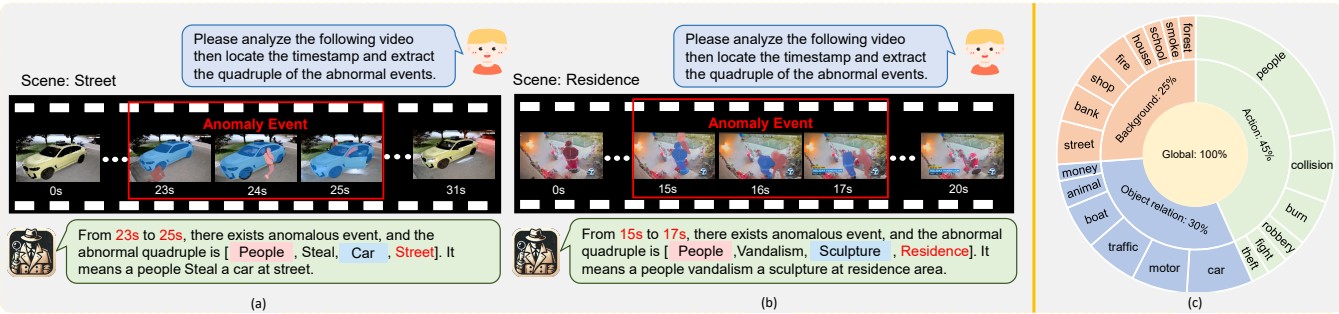

**Figure 1: (a) and (b) illustrate two surveillance video examples for our M-VAE task and Sherlock model in two scenes (Street and Residence). Sherlock precisely generates the abnormal event quadruples and their corresponding timestamps. (c) presents a circular ratio diagram illustrating different spatial information. From (c), we observe that the global spatial information and the local spatial information (i.e., action, object relation, and background) in our M-VAE dataset are imbalanced.**

## Abstract

In the literature, prior studies on Video Anomaly Detection (VAD) mainly focus on detecting whether each video frame is abnormal or not in the video, which largely ignore the structured video semantic information (i.e., what, when, and where does the abnormal event happen), though this structured information could be employed to construct a more precise and efficient system for abnormal event monitoring and retrieval. With this in mind, we propose a new chat-paradigm **M**ulti-scene **V**ideo **A**bnormal **E**vent Extraction and Localization (M-VAE) task, aiming to extract the abnormal event quadruples (i.e., subject, event type, object, scene) and localize such event. Further, this paper believes that this new task faces two key challenges, i.e., global-local spatial modeling and global-local spatial balancing. To this end, this paper proposes a Global-local Spatial-sensitive Large Language Model (LLM) named Sherlock, i.e., acting like *Sherlock Holmes* to track down the criminal events, for this M-VAE task. Specifically, this model designs a Global-local Spatial-enhanced MoE (GSM) module and a Spatial Imbalance Regulator (SIR) to address the above two challenges respectively. Extensive experiments on our constructed M-VAE instruction dataset show the significant advantages of Sherlock over several advanced Video-LLMs. This justifies the importance of global-local spatial information for the M-VAE task and the effectiveness of Sherlock in capturing such information.

## CCS Concepts

• **Computing methodologies → Artificial intelligence**.

## Keywords

Multi-scene Video, Video Abnormal Event, Spatial-sensitive LLM

## 1 Introduction

Video Understanding is a foundational task in artificial intelligence, which focuses on analyzing and interpreting the content of videos to enable various applications, including video classification, activity recognition, and scene understanding [42, 66, 67]. As a critical branch of video understanding, **V**ideo **A**nomaly **D**etection (VAD) [22], which aims to automatically detect abnormal videos, has garnered significant research attention due to its wide range of applications in criminal activity detection and disaster response [63]. Prior studies on VAD mainly focus on detecting whether each video frame is abnormal or not in the video [22, 31, 43, 63]. However, these studies overlook targeting at determining the underlying video semantic structure, i.e., "*what is the abnormal type, where they have occurred, which people or things are involved*" with a given video.

Motivated by these, this paper proposes a novel **M**ulti-scene **V**ideo **A**bnormal **E**vent Extraction and **L**ocalization (M-VAE) task[1], aiming at localizing abnormal events (i.e., starting and ending times of the anomaly) and extracting event quadruples (i.e. [subject of the event, event type, object of the event, scene of the event]) through a chat paradigm. Take an example of *Street* scene in Figure 1 (a), within 23s to 25s, a man bends down and pries the lock, then drives away from the street and the abnormal event quadruple is [*people, steal, car, street*]. Different scene (i.e., Residence scene) is also shown in Figure 1 (b). Within 15s to 17s, a man vandalizes a sculpture at one's residence and the quadruple is [*people, Vandalism, Sculpture, Residence*]. This structured processing for abnormal videos can significantly improve the practicality and efficiency of video anomaly localization systems. In fields such as real-time abnormal event monitoring that require high reliability and precision monitoring, using such structured processing can quickly search and screen for

---

[1]**Relevance to the Web**: M-VAE task belongs to *Development of structured data* topic of *Semantics and Knowledge* track, which aims to extract and locate quadruple from web videos (like YouTube and Tik Tok), making web content more accessible through video quick retrieval.

 

the required abnormal elements, which provides more convenient and intuitive evidence for further processing. Therefore, it is worthwhile to address this new task. Nevertheless, we believe that this new task faces two key challenges.

For one thing, it is challenging to model the global-local spatial information (named global-local spatial modeling challenge). Existing video understanding models [36, 40, 61] mainly focus on modeling general global information. However, local spatial information in our M-VAE task is often crucial compared to general global information, which are highly discriminative and essential for precise identification. Taking Figure 1 (a) as an example, the local spatial information, such as action (bend down), object relations (<man, near, car>), and background (street), can help better identify abnormal events. However, those local spatial information (e.g., actions, object relations, backgrounds) have different heterogeneous representations (i.e., different model structures and encoders). Therefore, a single, fixed-capacity transformer-based model, often makes it difficult to capture those critical local spatial information in videos. Recently, the Mixture of Expert (MoE) [20, 25] paradigm has demonstrated scalability in multi-modal heterogeneous representation fusion tasks [20, 25, 26]. Inspired by this, a well-behaved model for our task should adopt the MoE paradigm to not only consider global spatial information but also emphasize the importance of local spatial information.

For another, a straightforward approach is to employ a basic Mixture of Expert (MoE) mechanism [20, 25, 26] to treat global spatial information (i.e., general representations of videos) and local spatial information (e.g., actions) as the global expert and local experts for integrating those information. However, the data imbalance issue among local spatial information may lead to the basic MoE experts being biased towards the more frequently occurring spatial information in the dataset. The statistics in Figure 1 (c) can illustrate this imbalance. Certain frequently appearing local information (i.e., action at 45%), can lead to higher weight for the corresponding expert. However, in Figure 1 (a), the object relations information, with the smallest proportion (25%), but is the most discriminative for extracting and localizing *Theft* events. More seriously, global spatial information is the most frequent and our preliminary experiments in Figure 7 (a) reveal global expert is often more thoroughly trained and often have the highest weights. Therefore, a better-behaved MoE expert fusion mechanism should mitigate this data imbalance (named global-local spatial balancing challenge), ensuring all experts are sufficiently trained to highlight their importance.

To tackle above challenges, we propose a Global-local Spatial-sensitive LLM named Sherlock, i.e., acting like *Sherlock Holmes* to track down criminal events, for M-VAE. Specifically, this model designs a Global-local Spatial-enhanced MoE (GSM) module to address the global-local spatial modeling challenge, which includes four spatial experts to extract spatial information and an expert gate to weigh global and local spatial information. Furthermore, this model designs a Spatial Imbalance Regulator (SIR) to address the global-local spatial balancing challenge, which includes a Gated Spatial Balancing Loss (GSB) to further balance global and local experts. Particularly, we construct a M-VAE instruction dataset to better evaluate the effectiveness of our model. Detailed experiments show Sherlock can effectively extract and localize abnormal events and surpass advanced Video-LLMs in multiple evaluation metrics.

## 2 Related Work

• **Video Anomaly Detection**. Video Understanding is a rapidly evolving research field which encompasses several tasks, including video grounding [42, 66, 67], spatial-temporal detection [15] and so on. As an important branch of video understanding, previous studies on Video Anomaly Detection (VAD) can be categorized into unsupervised, weakly-supervised, and fully-supervised categories. Unsupervised approaches focus on leveraging reconstruction techniques to identify anomalies [17, 22, 71, 73]. Weakly-supervised methods have shown promising results in identifying abnormal frames [13, 38, 64, 68, 83]. Fully-supervised methods are scarce due to the expensive frame-level annotations required [9, 12, 14, 21, 60, 62, 80]. Different from the above studies, our Sherlock model aims to target at determining the underlying video semantic structure, providing a structured quadruple that goes beyond previous methods, facilitating the rapid detection and early warning of abnormal events in real-time.

• **Event Extraction** (EE) focuses on extracting structured information from given types of information. Traditional EE methods mainly extract from text documents [23, 27, 37, 39, 59]. Recently, many studies [3, 46, 75, 77, 78] generate similar event structures from visual image data. Different from all the above studies, we are the first to focus on extracting the abnormal event from videos and constructing a quadruple dataset, incorporating information from multiple spatial information, enriching the task of event extraction, and making it more practical for real-world applications.

• **Scene Recognition** is a fundamental task applied in remote sensing [8, 58] and autonomous driving [70]. Traditional methods rely on hand-crafted features for extracting visual attributes [8, 58]. Recently, ARCNet [65] and CapsNet [81] reinforcement, aim to locate important regions. Others, like using CapsNet in [2] and FACNN [48], focus on modeling global context. SCViT [50] and KFB combine fine-grained information. Recently, many studies [52, 55, 76] utilize LLMs to solve the illusion problem. Different from the above studies, we introduce scene classification into our M-VAE task and integrate scenes into event quadruples, greatly improving the applicability of our M-VAE task in the real world.

• **Video-oriented Large Language Models**. The rise of Chat-GPT [54] has stimulated the prosperity of Video Large Language Models which can be categorized into four major types: firstly, Video Chat [36] and Video LLaMA [79], which utilize BLIP-2 [35] and Q-Former to map visual representations onto Vicuna; secondly, models like Video ChatGPT [51], Otter [33], Valley [49], mPLUG-Owl [74], and Chat-UniVi [28], which leverage CLIP [57] to encode visual features; thirdly, PandaGPT [61], which adopts Image-Bind [16] as its core architecture for video understanding; and fourthly, VideoLLaVA [40], which aligns image and video features into a linguistic feature space using LanguageBind [84]. Recently, a few studies [29, 72] consider incorporating spatial information in models. Besides, some studies [20, 25, 26] introduce the concept of MoE into LLMs, but they only focus on efficiency, without considering the balance between different information. Different from all the above studies, we design a new Sherlock model, to address our M-VAE task, which includes a Global-local Spatial-enhanced MoE module and a Spatial Imbalance Regulator to address the challenges of global-local modeling and balancing.

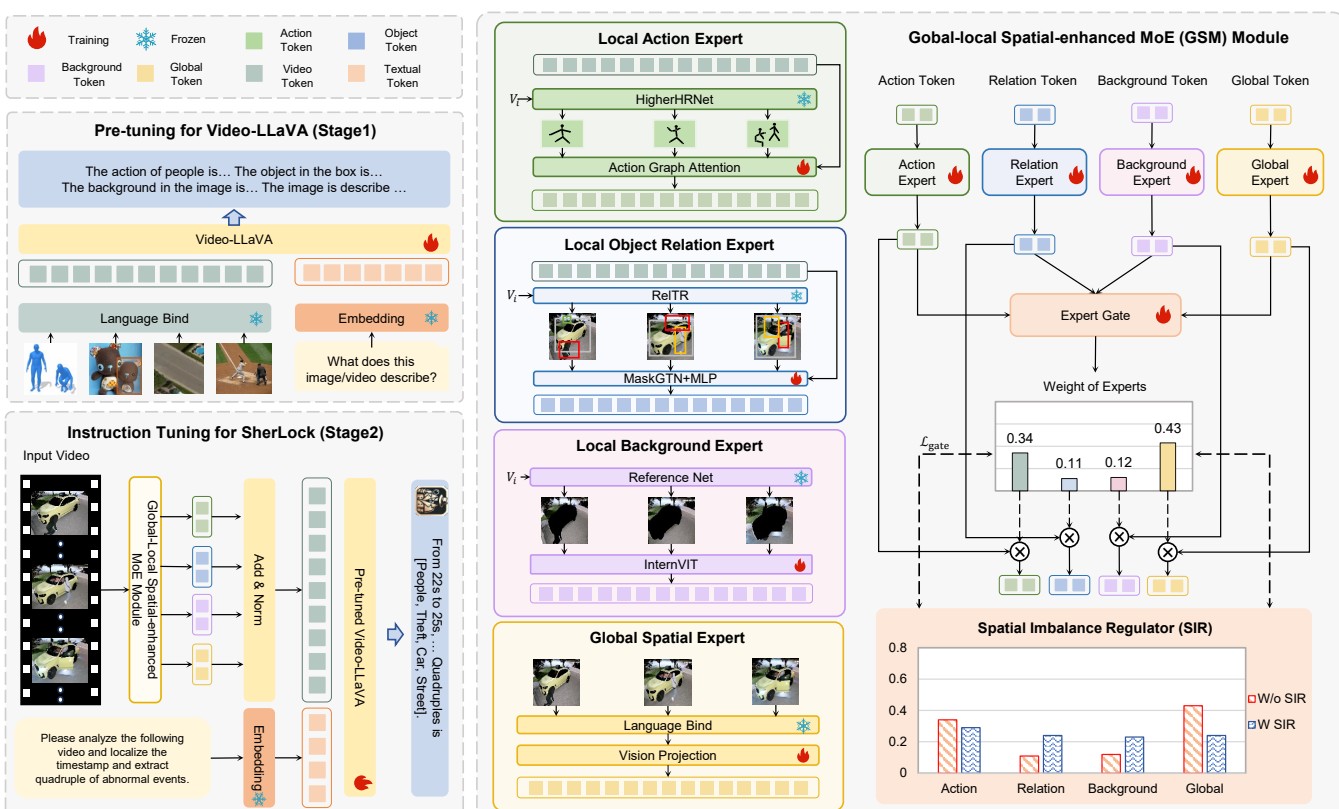

**Figure 2: The overall framework of Sherlock. It consists of a Global-local Spatial-enhanced MoE (GSM) Module and a Spatial Imbalance Regulator (SIR). The SIR exerts a direct influence on the output weights of the expert gate. W SIR or W/o SIR means with or without Spatial Imbalance Regulator.**

## 3 Our Sherlock Model

In this paper, we propose a Sherlock model to address the M-VAE task. Figure 2 illustrates the framework of Sherlock, which is composed of two core components (i.e., the Global-local Spatial-enhanced MoE (GSM) module (sec 3.1) for the global-local spatial modeling challenge and the Spatial Imbalance Regulator (SIR) (sec 3.2) for the global and local spatial balancing challenge). Subsequently, we present our training strategies to enhance the ability of understanding spatial information (sec 3.3).

**Backbone.** We choose Video-LLaVA[2] [40] and its visual encoder LanguageBind [84] as the core framework. Video-LLaVA, which is optimized with a mixed dataset of images and videos, demonstrates leading performance across most image and video benchmarks. We employ Video-LLaVA as the backbone to explore the potential of Video-LLMs in extracting and localizing abnormal events.

**Task Formulation.** Given a video $V$ for $M$ frames, each frame is labeled with 1 or 0, where 1 and 0 represent whether this frame conveys an abnormal event. The goal of M-VAE is to interactively generate the quadruple ($sub$, $type$, $obj$, $sce$) for each event along with the corresponding timestamp $sta$ and $end$, where $sub$, $type$, $obj$, $sta$ and $end$ are the subject, event type, object, scene, start time and end time of the abnormal event. As shown in Figure 1 (a), a man steals a car at street from 23s to 25s. Therefore, the output of our M-VAE task is {$23s$, $25s$, ($people$, $steal$, $car$, $street$)}.

---

[2]https://github.com/PKU-YuanGroup/Video-LLaVA.git

## 3.1 Global-local Spatial-enhanced MoE Module

As shown in Figure 2, we design a Global-local Spatial-enhanced MoE (GSM) Module for the global-local spatial modeling challenge. Inspired by Mixture-of-Experts (MoE) [26], we design three Local Spatial Experts (i.e., Local Action Expert, Local Object Relation Expert and Local Background Expert) and a Global Spatial Expert to extract spatial information, detailed as follows.

**Local Spatial Experts** contain three local spatial experts (i.e., action, object relation, and background), detailed as follows.

• **Local Action Expert (Action Expert, AE)**. We leverage HigherHRNet [7], a well-adopted bottom-up human pose estimation network to extract local spatial action information. HigherHRNet can generate local spatial action tokens $T_a = \{t_1^a, ..., t_i^a, ..., t_m^a\}$, and each token consists of 17 human joint nodes for each individual in every frame of a video sequence. Here, $i$ denotes the $i$-th frame. Next, we apply Action Graph Attention to integrate $T_a$ with the video tokens $T_v = \{t_1^v, ..., t_i^v, ..., t_m^v\}$ generated by the Video Encoder in Video-LLMs. We start by calculating the attention weights $\alpha_{kj}$ for each node $e_k$ in $t_i^a$ relative to its neighboring node $e_j$:

$$\alpha_{kj} = \text{softmax}\left(\frac{(\mathbf{W_a}h_k) \cdot (\mathbf{W_a}h_j)}{\sqrt{d}}\right) \quad (1)$$

where $h_k$ and $h_j$ is the features of $e_k$ and $e_j$ respectively. $\mathbf{W_a}$ denote the learnable weight matrix, and $d$ is the feature dimension. Then we aggregate the feature $\hat{h}_k$ of node $e_k$: $\hat{h}_k = \sum_{j \in \mathcal{N}(e_k)} \alpha_{kj} \cdot h_j$, where $\mathcal{N}(e_k)$ is the neighboring nodes of $e_k$. Finally the feature of

$e_k$ is calculated by $h'_k = \text{ReLU}(\mathbf{W_k}[\hat{h}_k, h_k])$, where $\mathbf{W_a}$ donates the weight matrix and $[\hat{h}_k, h_k]$ is the concatenation of $\hat{h}_k$ and $h_k$.

After graph attention operation, we enhance $\mathbf{T_a}$ using the attention mechanism with query $\mathbf{Q_v}$, key $\mathbf{K_a}$, and value $\mathbf{V_a}$ calculation to obtain final action tokens: $\mathbf{T'_a} = \text{softmax}\left(\mathbf{Q_v}^\top \cdot \mathbf{K_a}\right) \cdot \mathbf{V_a}$.

• **Local Object Relation Expert (Object Relation Expert, ORE)**. We leverage RelTR [11], a well-studied one-stage object relation graph generation method to extract local spatial object relation information. RelTR can generate an object relation token $t^o_i = (R_i, E_i)$, which represents the object relation graph of the $i$-th frame. Here, $R_i = \{(c_{i,1}, b_{i,1}), ..., (c_{i,k}, b_{i,k})\}$ is a set of $k$ detected objects, with class $c$ and corresponding bounding box $b$. The set $E_i = \{c_{i,p}, r_{i,(p,q)}, c_{i,q}\}$ consists of the directed edges in the graph, representing two directional edges from $c_{i,p}$ to $r_{i,(p,q)}$ and from $r_{i,(p,q)}$ to $c_{i,q}$, where $r_{i,(p,q)}$ denotes a relationship category. For example, an object might be represented as (*man, <0.36, 0.24, 0.75, 1.62>*), and an edge as (*man, near, car*). Subsequently, we apply object-aware masking with Masked Graph Transformer Networks (MaskGTN) to fully utilize object relations. We mask irrelevant object parts based on the bounding box information, and aggregate information from neighbors using a graph transformer layer (GT). Given an input graph of region classes and edges, MaskGTN computes updated vectors for each region and edge. Assuming we use $L$ layers of GT, with $\mathbf{H}^{(\ell)}$ representing the features of the $\ell$-th layer, the final forward propagation is defined as follows:

$$\mathbf{H}^{(\ell+1)} = \sigma\left(\sqrt{\tilde{\mathbf{D}}} \cdot \tilde{\mathbf{A}} \cdot \sqrt{\tilde{\mathbf{D}}} \cdot \mathbf{H}^{(\ell)} \cdot \mathbf{W}^{(\ell)}\right) \tag{2}$$

where $\sigma$ is the activation function on the graph. $\tilde{\mathbf{A}}$ is the adjacency matrix of the object-relation graph, derived from $E_i$, and $\tilde{\mathbf{D}}$ is its degree matrix, with $\tilde{\mathbf{D}}_{ii} = \sum_i \tilde{\mathbf{A}}_{ij}$. $\mathbf{W}^{(\ell)}$ is a trainable weight matrix.

• **Local Background Expert (Background Expert, BE)**. We leverage SAM2 [30], an advanced model for visual segmentation, to extract local spatial background information from videos. SAM2 can generate a background image for each frame of video. Then we leverage InternVit [6] to encode local spatial background information which is a large vision encoder extending the parameters of vision transformer (VIT) [5] to 6B, formally represented as:

$$\mathbf{T_b} = \text{InternVit}(\text{SAM2}(v_i)) \tag{3}$$

where $v_i$ is the $i$-th frame of video $V$. This process results in the local spatial background tokens $\mathbf{T_b} = \{t^b_1, ..., t^b_i, ..., t^b_m\}$ for the entire video sequence, with $n$ representing the total number of frames.

**Global Spatial Expert** has a comprehensive understanding of the training data. Collaborate with local spatial experts to bring specialization and generalization capabilities to M-VAE tasks.

• **Global Spatial Expert (Global Expert, GE)**. The weight assigned to the global spatial expert complements that of the local spatial experts. Consequently, the local spatial experts acquire specialized skills for specific tasks, whereas the global spatial expert develops a comprehensive understanding of the entire training corpus. The collaboration between these two types of experts provides both specialization and generalization for our M-VAE task. In this way, we leverage LanguageBind [84] in Video-LLaVA [40], which inherits the ViT-L/14 structure from CLIP and is equipped with powerful and universal visual encoding capabilities to extract global spatial information for our task. We subsequently leverage a pre-trained FFN layer by [40] to align the dimension with other spatial information, formally represented as:

$$\mathbf{T_g} = \text{FFN}(\text{LanguageBind}(v_i)) \tag{4}$$

where $v_i$ is the $i$-th frame of video $V$. This process yields the full set of global tokens $\mathbf{T_g} = \{t^g_1, ..., t^g_i, ..., t^g_m\}$ for the entire video sequence, with $n$ representing the total number of frames.

After designing four experts, we ensure that the four Spatial Experts can dynamically adjust the weights of the four heterogeneous types of spatial information inspired by Mixture-of-Experts (MoE) [20]. As shown in Figure 2, unlike methods that embed several FFNs within LLMs, our GSM put four experts outside the LLMs to adjust weights for global and local spatial information. Based on this, we introduce a dynamic Expert Gate (EG) [56], which controls the contribution of each expert by calculating gating weights as a soft gate. Finally, the output $\mathbf{O}$ of GSM, based on four spatial experts and EG, is formally represented as:

$$g = \text{softmax}\left(\mathbf{W}_g \cdot \sum_{i=1}^{N}(\mathbf{S_i})\right) \tag{5}$$

$$\mathbf{O} = \text{LayerNorm}\left(\sum_{i=1}^{N}(g_i \cdot \mathbf{S_i})\right) \tag{6}$$

where $\text{LayerNorm}(\cdot)$ indicates layer normalization [1]. $g_i$ (the $i$-th entry in $g$) represents the weight of the $i$-th expert. $\mathbf{S_i}$ represents the outputs of the $i$-th Spatial expert. $N$ is the total number of spatial expert, and $\mathbf{W}_g$ being the trainable weight matrix.

## 3.2 Spatial Imbalance Regulator

After modeling the spatial information, we design a Spatial Imbalance Regulator (SIR) including a Gated Spatial Balancing Loss (GSB) for the global-local spatial balancing challenge, detailed as follows.

**Gated Spatial Balancing (GSB) Loss.** Previous researches employ a basic Mixture of Experts (MoE) [20, 25] to model global and local spatial information. When faced with an imbalance between these two types of information, the weights assigned to experts tend to be biased toward those that appear more frequently. As shown in Figure 1 (c), there are the most spatial elements (e.g., *People*) related to local spatial action information in event quadruple. This implies that performance will deteriorate when faced with real-world data that is not processed by an action expert (e.g., *object relations*). More seriously, as shown in Figure 1 (c), global information holds significant weight in all data, which will lead to excessive training of global experts and weaken the abilities of local experts with lower weights. This imbalance phenomenon will greatly affect the performance of our model. Based on this, we should keep the weights of all spatial experts not too different and achieve the optimal state of relative balance where every expert is fully trained. Inspired by MoELoRA [44], we propose a **G**ated **S**patial **B**alancing (GSB) Loss to balance spatial weights, as follows:

$$\mathcal{L}_{\text{gate}} = \left(\frac{1}{N_{\text{local}}}\sum_{i=1}^{N_{\text{local}}} -\log(g_i)\right) - \log\left(g_{\text{global}}\right) \tag{7}$$

**Table 1: The statistics of the number of events and the duration in seconds (s) of events for each scene.**

| Split | School | Shop | Underwater | Street | Road | Boat | Wild | Forest | Residence | Bank | Commercial | Factory | Lawn | Other | Total |
|---|---|---|---|---|---|---|---|---|---|---|---|---|---|---|---|
| Train | 55 (2136s) | 107 (4130s) | 78 (3022s) | 113 (7076s) | 114 (5586s) | 115 (5203s) | 111 (4681s) | 102 (3918s) | 117 (4914s) | 89 (3380s) | 105 (5011s) | 82 (3173s) | 104 (5943s) | 56 (1497s) | 1348 (59670s) |
| Inference | 13 (534s) | 26 (1032s) | 19 (755s) | 28 (1769s) | 28 (1396s) | 29 (1300s) | 27 (1170s) | 25 (979s) | 29 (1228s) | 22 (845s) | 26 (1252s) | 20 (793s) | 26 (1485s) | 14 (374s) | 332 (14912s) |

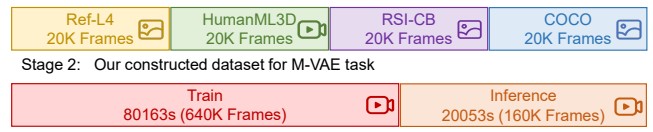

**Stage 1:** The dataset of pre-tuning for spatial understanding

Ref-L4 20K Frames    HumanML3D 20K Frames    RSI-CB 20K Frames    COCO 20K Frames

**Stage 2:** Our constructed dataset for M-VAE task

Train 80163s (640K Frames)    Inference 20053s (160K Frames)

**Figure 3: Data composition for training and inference.**

where $N_{\text{local}}$ is the number of local expert. $g_{\text{global}}$ is the weight of global expert. The first term of Eq.(7) is balancing between local experts, and the second term is balancing between local and global experts. The weights of four experts have already balanced when the loss is optimized to a minimum. This regulation achieves a better balance among all experts, reducing the impact of data imbalance, which effectively addresses the global-local balancing challenge. Finally, the overall loss of Sherlock can be represented as:

$$\mathcal{L} = \mathcal{L}_{\mathcal{D}} + \alpha * \mathcal{L}_{\text{gate}} \tag{8}$$

where $\alpha$ is the hyper-parameter that controls the strength of $\mathcal{L}_{\text{gate}}$, and $\mathcal{L}_{\mathcal{D}}$ is the next-token prediction loss of Video-LLMs.

### 3.3 Training Strategies for Sherlock

In order to enhance the ability of understanding spatial information, we design a two-stage training process. Stage 1 is to enhance the ability of understanding spatial information and Stage 2 is to address the M-VAE task, detailed as follows.

**Stage 1. Pre-Tuning for spatial understanding.** As shown in Figure 2, we first pre-tune Video-LLaVA using four high-quality datasets. We aim for Video-LLaVA to have a good spatial understanding ability. Specifically, we selected four high-quality datasets: HumanML3D [18], Ref-L4 [4], RSI-CB [34], and COCO-Caption [41], as described in sec 4.1. For each pre-tuning dataset, we enable this dataset to understand corresponding spatial information.

**Stage 2. Instruction Tuning for M-VAE task.** We aim to enable the model to localize abnormal events and extract quadruples through the chat paradigm. We construct an instruction tuning dataset described in sec 4.1 and instruct the pre-tuned Video-LLaVA to *Extract quadruples and localize abnormal events. The quadruple includes subject, event type, object, and scene in abnormal events.* The instruction will undergo text embedding to obtain the textual tokens $\mathbf{T}_t$. Finally, the input of the LLM is "$\mathbf{O}$ from Eq.(5) + $\mathbf{T}_t$".

## 4 Experimental Settings

### 4.1 Instruction Data Construction

The training pipeline of Sherlock contains two stages. As shown in Figure 3, for each stage, we construct the corresponding instruction dataset for better tuning.

**For Stage 1.** We construct a special understanding dataset based on Ref-L4 [4], HumanML3D [18], RSI-CB [34] and COCO [41]. Specifically, we manually design an instruction for each type of spatial information, for instance: **Instruction**: "*Judge the action of the characters in the image. Describe the image region <objs> in the image. Judge the background of the image. Describe the image*". As HumanML3D has 25K videos with an average duration of 1 second,

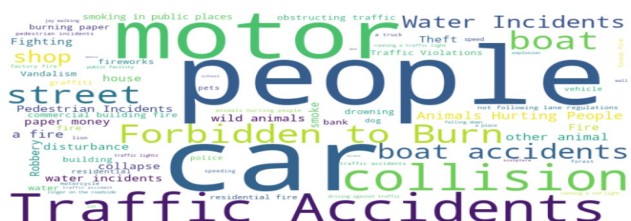

**Figure 4: The word cloud distribution of quadruple elements in the M-VAE dataset, which reveals the spatial imbalance. (e.g., The proportion of *people* is the highest)**

and we take 8 frames per second. For the data balance, we randomly select 20K images or frames from each dataset.

**For Stage 2.** We construct an M-VAE instruction dataset based on CUVA [12], which primarily consists of surveillance videos, with an average duration of **80** seconds per video. As this dataset includes five detailed video Q-A tasks (i.e., timestamp, classification, reason, result, and description tasks), it is highly beneficial for constructing our M-VAE dataset. **1)** For abnormal event quadruples, constructing quadruples involves two steps. **First**, we collect answers from the reason, result, and description tasks in CUVA for each video. Subsequently, we construct initial quadruples through ChatGPT [54] based on the answers to these tasks, with the instruction: "*Please extract the subject, object, and scene of the event based on the responses below*". **Second**, we create multiple candidate sets for subjects, objects, and scenes in quadruple. Specifically, **for subjects and objects elements**, we manually construct a set of around 40 for subjects and objects and filter elements based on this set. **For event types elements**, we adopt the 11 categories (i.e., Fighting, Animals, Water, Vandalism, Accidents, Robbery, Theft, Pedestrian, Fire, Violations, and Forbidden) from CUVA as the event types. **For scenes elements**, we assign two annotators to classify scenes for each abnormal event. If they cannot reach an agreement, an expert will make the final decision to ensure annotation quality. The *Kappa* consistency check value of the annotation is 0.87. **2)** For localization task, we use the timestamp in the CUVA as labels for localization. Furthermore, we adhere to the split of CUVA for training and inference videos and take 8 frames per second, resulting in **800K** frames from 1k videos and each video contains **1.68** abnormal event on average. The statistics of the number of events and the duration in seconds (s) of events for each scene are shown in Table 1. Finally, we obtain our M-VAE instruction dataset. Our instruction for the M-VAE task is: "*Generate a quadruple and localize an abnormal event in the video. The quadruple includes subject, event type, object, and scene in abnormal events.*". Figure 1 (c) and Figure 4 show the top 20 quadruple elements, revealing the spatial imbalance.

### 4.2 Baselines

In this paper, we select several advanced Video-LLMs as baselines which are introduced as follows. **VideoChat** [51] employs Q-Former [35] to map visual representations to Vicuna [10]. **Video-ChatGPT** [51] integrates LLMs with CLIP [57] for video representations. **Valley** [49] employs a temporal modeling module to bridge

Table 2: Comparison of several Video-LLMs and Sherlock on our instruction dataset, wherein evaluation for Anomaly Cls. (i.e., Anomaly classification) is to assess traditional anomaly classification task [19, 47, 69] (i.e., whether each video frame is ab- normal or not in the video). The ↓ beside FNRs indicates the lower the metric, the better the performance. AE, ORE, BE, GE, and EG represent four Spatial Experts and Expert Gate respectively. Sub, Type, Obj, and Sce represent Subject, Event type, Object, and Scene respectively. T5-based and GPT-based metrics are based [53] for LLM especially. For each task, Blue and Green donate the first and second place respectively.

| Models | Event Extraction | | | | | | | | | | | | Event Location | | | | Anomaly Cls. | |
| | Single (F1) | | | | Pair (F1) | | | | Quadruple | | | Average | mAP@tIoU | | | Average | FNRs | F2 |
| | Subject | Type | Object | Scene | Sub-Type | Obj-Type | Sub-Sce | Obj-Sce | F1 | T5-based | GPT-based | | 0.1 | 0.2 | 0.3 | | | |
|---|---|---|---|---|---|---|---|---|---|---|---|---|---|---|---|---|---|---|
| Video Chat | 73.14 | 71.35 | 64.28 | 71.76 | 70.12 | 58.69 | 71.55 | 61.18 | 40.95 | 51.68 | 53.94 | 62.6 | 77.28 | 74.93 | 66.26 | 72.82 | 38.79 | 65.88 |
| Video ChatGPT | 61.87 | 59.51 | 54.82 | 46.39 | 54.23 | 49.68 | 43.26 | 41.38 | 39.63 | 47.36 | 50.38 | 49.86 | 74.65 | 70.91 | 67.03 | 70.86 | 41.47 | 61.35 |
| Valley | 64.64 | 62.27 | 58.94 | 52.26 | 58.36 | 51.64 | 49.68 | 46.42 | 42.38 | 53.34 | 56.67 | 54.23 | 69.34 | 62.26 | 57.66 | 63.08 | 43.49 | 59.42 |
| Panda GPT | 73.09 | 75.45 | 68.42 | 61.93 | 71.96 | 59.92 | 59.79 | 59.45 | 41.17 | 44.36 | 48.55 | 60.37 | 76.64 | 62.69 | 57.21 | 65.51 | 35.62 | 69.16 |
| mPLUG-Owl | 52.86 | 37.54 | 40.24 | 37.68 | 31.97 | 28.89 | 33.9 | 27.87 | 22.12 | 29.68 | 32.41 | 34.1 | 61.42 | 53.21 | 46.46 | 53.69 | 56.98 | 51.66 |
| Chat-UniVi | 59.71 | 57.26 | 55.28 | 44.23 | 52.43 | 50.62 | 41.24 | 40.96 | 37.68 | 45.34 | 48.84 | 43.59 | 65.89 | 58.62 | 40.02 | 54.84 | 52.52 | 53.78 |
| Video-LLaVA | 77.85 | 73.68 | 65.67 | 75.91 | 69.32 | 59.21 | 73.25 | 62.24 | 41.32 | 52.94 | 56.74 | 64.37 | 78.31 | 74.79 | 64.92 | 72.67 | 41.34 | 64.96 |
| Sherlock | 87.97 | 82.12 | 74.99 | 92.15 | 77.06 | 66.28 | 85.16 | 73.17 | 57.57 | 65.46 | 67.52 | 75.22 | 94.03 | 82.59 | 76.12 | 84.24 | 17.24 | 83.59 |
| w/o AE | 83.15 | 77.64 | 71.28 | 90.16 | 72.36 | 63.47 | 80.52 | 70.39 | 52.48 | 59.61 | 62.02 | 71.18 | 92.24 | 81.21 | 75.38 | 82.94 | 21.82 | 80.45 |
| w/o ORE | 83.96 | 78.25 | 72.37 | 90.01 | 74.24 | 64.46 | 81.56 | 70.97 | 54.35 | 62.28 | 65.08 | 72.5 | 91.13 | 82.08 | 74.62 | 82.61 | 22.97 | 78.83 |
| w/o BE | 81.16 | 74.65 | 67.88 | 88.07 | 69.29 | 61.12 | 77.64 | 66.64 | 48.63 | 53.04 | 55.94 | 67.71 | 88.62 | 79.09 | 72.24 | 79.98 | 25.36 | 73.51 |
| w/o GE | 79.2 | 74.09 | 66.71 | 84.11 | 70.38 | 60.77 | 75.44 | 66.28 | 46.34 | 53.97 | 57.06 | 66.75 | 86.18 | 78.37 | 69.28 | 77.94 | 28.97 | 71.28 |
| w/o EG | 78.83 | 73.96 | 65.02 | 83.15 | 70.15 | 60.26 | 74.15 | 63.37 | 43.64 | 49.14 | 51.82 | 64.86 | 81.31 | 77.68 | 67.88 | 75.62 | 32.58 | 67.07 |
| w/o SIR | 84.47 | 80.14 | 71.94 | 92.34 | 75.58 | 64.84 | 83.21 | 70.06 | 55.73 | 62.87 | 65.18 | 73.3 | 83.41 | 78.49 | 68.37 | 76.75 | 30.64 | 70.97 |
| w/o pre-tuning | 78.24 | 74.44 | 64.22 | 82.21 | 68.55 | 57.74 | 72.62 | 62.91 | 42.51 | 47.22 | 50.54 | 63.74 | 79.58 | 75.32 | 65.07 | 73.32 | 34.87 | 66.64 |

visual and textual modes. **PandaGPT** [61] utilizes ImageBind [16] to demonstrate cross-modal capabilities. **mPLUG-Owl** [74] introduces a visual abstractor module to align different modes. **Chat-UniVi** [28] merges visual tokens with semantic meanings. **Video-LLaVA** [40] conducts joint training on images and videos. To ensure a fair comparison, we re-implement these models using their released codes in our experiments, with all LLMs sized at 7B.

### 4.3 Evaluation Metrics

M-VAE focuses on extracting event quadruples and locating abnormal events from videos, requiring evaluation metrics in three aspects (i.e., extract event quadruples, locate abnormal events, and classify abnormal events). **For the extraction performance**, we measure our model through three perspectives. **1)** Single: performance of generating each individual element. **2)** Pair: performance of generating the element pair, i.e., Subject-Type pair, Object-Type pair, Subject-Scene pair, Object-Scene pair. **3)** Quadruple Generation: performance of generating the complete event quadruple. Following the prior works [32], the performance is evaluated with Macro-F1. Furthermore, we use T5-based and GPT-based metrics based on Video-bench [53] especially for LLM. **For localization performance**, we use the mAP@tIoU metric [82], calculated by mean Average Precision (mAP) at different IoU thresholds from 0.1 to 0.3 with 0.1 intervals. **For classification performance**, we refer to the traditional anomaly classification task [19, 47, 69] for anomaly classification metric, which mainly determines whether each video frame is abnormal or not in the video. We prefer Recall over Precision and report F2 [82] as another classification metric. Furthermore, our model focuses on accurately distinguishing abnormal events. As shown in Figure 1, it's better to mark all timestamps as abnormal than to miss any. So we prioritize false negative rates (FNRs): FNRs = $\frac{num\ of\ false\text{-}negative\ frame}{num\ of\ positive\ frame}$, which is the rate of mislabeling an abnormal event frame as normal. In addition, $t$ test[3] is used to evaluate the significance of the performance.

[3]https://docs.scipy.org/doc/scipy/reference/stats.html

### 4.4 Implementation Details

In our experiments, we utilize open-source codes to obtain experimental results of all the baselines in Table 2 and Table 4. The hyper-parameters of these baselines remain the same setting reported by their public papers. The others are tuned according to the best performance. For both Stage 1 and 2, we use a batch size of 16 and train for 1 epoch with the AdamW [45] optimizer and a cosine learning rate decay schedule with a warm-up period. The initial learning rate is 2e-5. The hyper-parameter $\alpha$ in $\mathcal{L}$ is set to 0.4. We tune the Video-LLaVA model using LoRA [24]. The LoRA matrix dimension, dropout rate, and dropout rate are 16, 64, and 0.05 respectively. Experiments are run on a single NVIDIA A100 GPU with 40GB memory. Stage 1 training takes about 16 hours, Stage 2 takes 60 hours, and inference takes about 8 hours. To facilitate the corresponding research, all codes and the M-VAE datasets will be released via GitHub.

## 5 Results and Discussions

### 5.1 Experimental Results

Table 2 shows the performance comparison of different models on our M-VAE task, and we can see that: **For extraction performance**, our **Sherlock** model outperforms all baselines, with an average improvement of 10.85 ($p$-value < 0.05) over the second-best performance. More specifically, our **Sherlock** model surpasses the second-best performance by an average of 9.9 ($p$-value < 0.05), 8.59 ($p$-value < 0.05), and 9.52 ($p$-value < 0.05) in average Single, Pair, and Quadruple metrics, justifying the effectiveness of our **Sherlock** model on extraction task. **For localization performance**, our **Sherlock** model exceeds the second-best performance by 11.42 ($p$-value < 0.01) in average mAP@tIoU metric, justifying the effectiveness of our **Sherlock** model on localization task. Furthermore, **for classification performance**, in FNRs and F2 metric, our **Sherlock** model surpasses the second-best performance in 18.38 ($p$-value < 0.01) and 14.43 ($p$-value < 0.01). This implies the importance of our global and local spatial information and justifies the effectiveness of our **Sherlock** model on classification task.

**Table 3: Comparison of several advanced Video-LLMs and Sherlock on the 14 scenes of the M-VAE dataset with FNRs.**

| Models | School | Shop | Underwater | Street | Road | Boat | Wild | Forest | Residence | Bank | Commercial | Factory | Lawn | Other |
|--------|--------|------|-----------|--------|------|------|------|--------|-----------|------|------------|---------|------|-------|
| Video Chat | 39.57 | 39.47 | 37.3 | 36.81 | 27.41 | 35.32 | 33.27 | 33.36 | 35.95 | 40.59 | 38.97 | 45.52 | 35.26 | 49.04 |
| Video Chatgpt | 45.91 | 41.98 | 39.36 | 41.41 | 30.11 | 38.19 | 36.32 | 37.73 | 37.54 | 44.5 | 42.96 | 40.78 | 36.28 | 52.33 |
| Valley | 46.68 | 43.76 | 41.37 | 44.24 | 35.66 | 42.15 | 46.78 | 39.25 | 42.15 | 48.35 | 48.31 | 47.21 | 37.11 | 53.09 |
| Pandagpt | 34.56 | 35.65 | 34.47 | 36.48 | 24.42 | 35.85 | 31.78 | 32.37 | 34.18 | 38.55 | 37.89 | 41.46 | 31.17 | 44.24 |
| mPLUG-Owl | 54.13 | 54.41 | 53.21 | 47.34 | 36.51 | 45.02 | 58.37 | 46.31 | 45.63 | 57.94 | 56.88 | 53.14 | 54.74 | 59.56 |
| Chatunivi | 52.51 | 48.82 | 47.52 | 48.68 | 35.53 | 44.41 | 59.88 | 45.96 | 44.34 | 54.92 | 55.66 | 51.12 | 52.22 | 55.48 |
| Video-llava | 45.27 | 37.43 | 34.63 | 38.84 | 27.76 | 32.54 | 26.41 | 30.29 | 31.45 | 21.19 | 29.84 | 20.08 | 30.72 | 28.31 |
| **Sherlock** | **16.35** | **21.91** | **15.16** | **24.24** | **14.63** | **20.96** | **17.29** | **18.48** | **20.43** | **11.21** | **23.43** | **8.96** | **21.44** | **13.6** |

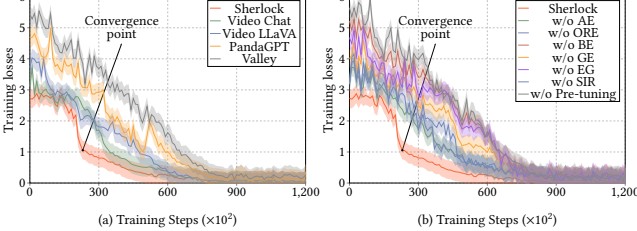

(a) Training Steps (×10²)    (b) Training Steps (×10²)

**Figure 5: Convergence analysis of other baselines, Sherlock, and its variant without specific components.**

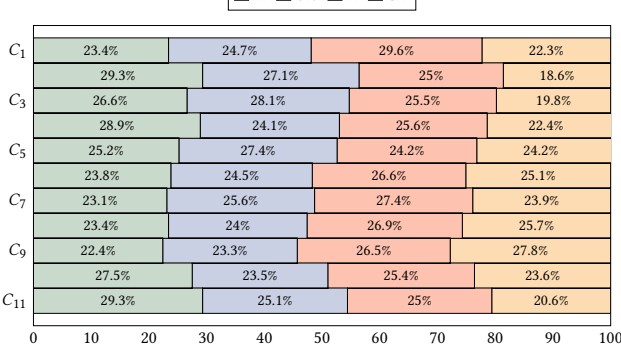

**Figure 6: The visualization of balanced spatial expert weights calculated in Eq.(5). The length of the bar in different colors represents the weights for the corresponding expert. $C_1$ to $C_{11}$ is different Event types in quadruples.**

## 5.2 Contributions of Each Key Component

In order to further investigate the contributions of different modules of **Sherlock**, we conduct an ablation study on our **Sherlock** model. As shown in Table 2, w/o AE, w/o ORE, w/o BE, w/o GE, w/o EG, and w/o pre-tuning represent without four Spatial Experts, Expert Gate, and pre-tuning stage in sec 3.2 respectively.

**Effectiveness Study of Global and Local Spatial Expert**. From Table 2, we can see that: The performance of **w/o AE**, **w/o ORE**, **w/o BE** and **w/o GE** degrades in all metrics, with an average decrease of 7.54 ($p$-value < 0.01), 7.57 ($p$-value < 0.01), 4.37 ($p$-value < 0.01), and 5.68 ($p$-value < 0.01) in FNRs, F2, average map@tIoU, and average event extraction metrics. This confirms the importance of global and local spatial information in extracting and localizing abnormal events, and our **Sherlock** model can better model those information well.

**Effectiveness Study of Spatial Imbalance Regulator.** From Table 2, we can see that: **1)** Compared with **Sherlock**, **w/o EG** shows poorer performance in all metrics, with a decrease of FNRs, F2, average map@tIoU, and average extraction performance by

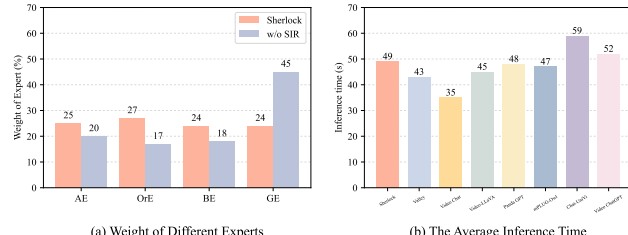

(a) Weight of Different Experts    (b) The Average Inference Time

**Figure 7: (a) is the visual comparison of our SIR and (b) is the comparison of the average inference time for a one-minute video between Sherlock and other Video-LLMs.**

**Table 4: Comparison of localization and anomaly classification task with several well-performing non-LLM models which is conducted on publicly available datasets.**

| Models | Anomaly Location | | | | Anomaly Cls. | |
|--------|------|------|------|---------|------|------|
| | mAP@tIoU | | | Average | FNRs | F2 |
| | 0.1 | 0.2 | 0.3 | | | |
| BiConvLSTM[21] | 52.74 | 37.31 | 31.12 | 40.39 | 68.05 | 44.48 |
| SPIL[62] | 53.28 | 38.89 | 32.91 | 41.69 | 67.84 | 46.87 |
| FlowGatedNet[9] | 53.64 | 39.64 | 33.18 | 42.15 | 67.24 | 46.55 |
| X3D[60] | 54.52 | 40.05 | 34.96 | 43.17 | 65.08 | 48.65 |
| HSCD[14] | 56.14 | 42.87 | 35.28 | 44.76 | 60.36 | 52.28 |
| **Sherlock** | **94.03** | **82.59** | **76.12** | **84.24** | **17.24** | **83.59** |

15.34 ($p$-value < 0.01), 16.52 ($p$-value < 0.01), 8.62 ($p$-value < 0.05) and 10.36 ($p$-value < 0.01), respectively. This demonstrates the effectiveness of GSM in global-local spatial modeling and encourages us to consider handling heterogeneity issues between spatial information in the manner of MoE. **2)** From Table 2, we can see that compared to performance of **w/o SIR**, the performance of **w/o MG** is poorer, with FNRs, F2, average map@tIoU, and average event extraction metrics decreasing by 1.94 ($p$-value < 0.05), 3.9 ($p$-value < 0.05), 1.13 ($p$-value < 0.05) and 4.84 ($p$-value < 0.05), respectively. This further demonstrates the effectiveness of $\mathcal{L}_{gate}$ in global-local spatial balancing and encourages us to consider using SIR to better balance spatial information. **3)** In addition, we record the weights of four spatial experts after training in Figure 6 and Figure 7 (a). We can see that the weights of all experts have been relatively balanced, and each expert has demonstrated outstanding professional abilities when facing different types of abnormal videos.

**Effectiveness Study of Pre-tuning**. From Table 2, we can see that **w/o pre-tuning**, the performance is inferior to **Sherlock**. FNRs, F2, average map@tIoU, and average event extraction metrics have decreased by 17.63 ($p$-value < 0.01), 16.95 ($p$-value < 0.01), 10.92 ($p$-value < 0.01) and 11.48 ($p$-value < 0.01), respectively. This further

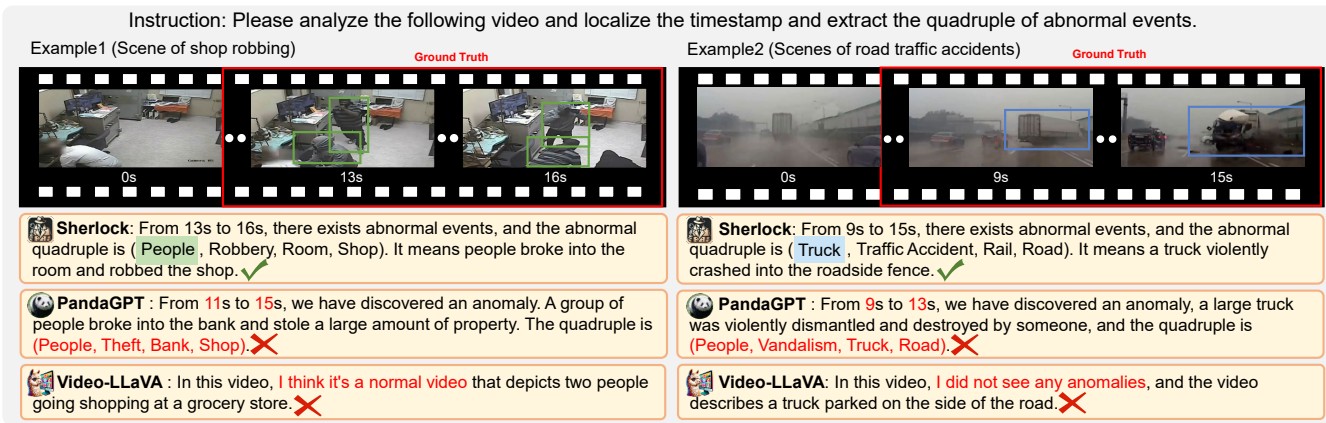

**Figure 8: Two Visualized samples to compare Sherlock with other Video-LLMs.**

justifies the effectiveness of pre-tuning, as well as encourages us to use more high-quality datasets to enhance the spatial understanding ability of Video-LLMs before instruction-tuning.

### 5.3 Convergence Analysis and Practical Assessment for Sherlock

In order to analyze the convergence of Sherlock, we record the loss of baseline Video-LLMs, Sherlock, and its variant without specific components over various training steps during the experiment. The results are shown in Figure 5 and we can see that: **1) Sherlock** demonstrates the fastest convergence compared to other Video-LLMs. At the convergence point, the loss of Sherlock is 1.05, while Video-LLaVA is 2.06. This underscores the high efficiency of Sherlock over other advanced Video-LLMs, which hints at the potential of Sherlock for quicker training steps and less resource utilization. **2) Sherlock** demonstrates the fastest convergence compared to its variant without specific components in Figure 5. This justifies that the four types of spatial information along with GSM and SIR can accelerate the convergence process, which further encourages us to consider the spatial information in the M-VAE task.

To assess practicality, we analyze the FNRs of Sherlock for each scene. As shown in Table 3, we can observe that in every scene, Sherlock outperforms other Video-LLMs. This indicates that the possibility of misclassifying abnormal events as normal events is minimized, thereby demonstrating the importance of global and local spatial modeling of Sherlock. We also analyze the average inference time in seconds for a one-minute video. As shown in Figure 7 (b), Sherlock does not perform much differently from the other models in terms of inference time. This is reasonable, as some studies confirm that the MoE architecture can improve efficiency [11, 28]. This suggests that introducing more information along with a MoE module for the M-VAE task does not increase the inference time and Sherlock can maintain good inference efficiency.

### 5.4 Compared with Advanced Non-LLM Models on Public Dataset

In order to more comprehensively evaluate the effectiveness of Sherlock, we compare our **Sherlock** model with other advanced non-LLM models [9, 14, 21, 60, 62] on traditional anomaly localization and anomaly classification task based on publicly available CUVA datasets [12]. Specifically, we need Sherlock to determine

whether each second of the video is abnormal or not without generating quadruples. As shown in Table 4, non-LLM models not only underperform relative to other Video-LLMs presented in Table 4 but also significantly inferior to our Sherlock model. This further demonstrates the importance of the global and local spatial information we proposed for the M-VAE task.

### 5.5 Qualitative Analysis for Sherlock

As shown in Figure 8, we visualize and compare **Sherlock** with other Video-LLMs. We randomly select two samples from our dataset and ask these models to *Analyze the following video and localize the timestamp and extract the quadruple of the abnormal events.* From the figure, we can see that: **1)** Accurately localizing abnormal events and extracting correct quadruples is a huge challenge. For instance, example 2 captures a segment from 9s to 15s, where identifying the collision of the truck at road is particularly challenging, **2)** Compared with other advanced Video-LLMs, **Sherlock** shows excellent performance in localizing abnormal events. In example 1, **Sherlock** outperforms other models in terms of prediction accuracy. In example 2, it outperforms PandaGPT in terms of accuracy and can generate a correct quadruple. This further demonstrates the effectiveness of **Sherlock** in precisely extracting and localizing abnormal events in video segments.

### 6 Conclusion

In this paper, we firstly propose a new M-VAE task and a constructed M-VAE instruction dataset, making a significant contribution to future research on abnormal events. Secondly, we propose a Global-local Spatial-sensitive LLM named Sherlock to assist in localizing and extracting abnormal event quadruples, providing decision-makers with more intuitive and comprehensive information support. This model includes a Global-local Spatial-enhanced MoE module and Spatial Imbalance Regular to model and balance spatial information. In the end, our experimental results demonstrate the outstanding performance of Sherlock. In future work, we hope to consider the relationships between events and enrich our tasks with event inference to improve the performance of extraction. In addition, we also hope to improve the interpretability of our model by providing explanations for each abnormal event.

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
