# OpenReview forum: "Sherlock: Towards Multi-scene Video Abnormal Event Extraction and Localization via a Global-local Spatial-sensitive LLM"
_ACM.org/TheWebConf/2025/Conference — WWW 2025 Poster_

### Official Review · Reviewer_mnYG · 2024-11-11

**Novelty:** 4
**Technical Quality:** 4

**Review:**

The author proposes a new chat paradigm, the Multi-scenario Video Anomaly Event Extraction and Localization Task (M-VAE), and a large language model for video anomaly detection named Sherlock.

Pros:
1. The author presents a good analysis of the current status and related work in the community.
2. The article is well-organized and easy to understand.

Cons:
1. The author attempts to cover too much content within a limited space, resulting in the proposed method and dataset not being comprehensively evaluated.
2. The model structure is overly complex and depends on too many different prerequisite models.
3. There is a lack of innovation in the method design.

**Questions:**

1. There are typo errors regarding capitalization in lines 104 and 106.
2. It should be $W _ k$ instead of $W _ a$ in line 349.
3. Given the complexity of the methods, a comparison of the time consumption and space usage among different methods should be added, especially with those non-LLM methods.
4. In Figure 5(b), it's difficult to distinguish the different curves. Not only are the curves too thin, but also their colors and overlaps make them hard to understand.
5. I've read Citation [44], but I don't understand why Eq. 7 can achieve weight balance. Judging from this formula, it can only ensure that multiple modules are optimizable, but not that they are balanced. I hope a more detailed explanation and proof analysis can be provided.
6. In Eq. 8, does the hyperparameter have a significant impact on the final model? Why is this hyperparameter set? What's even stranger is that I haven't found any sensitivity experiments or other analytical experiments regarding it. What impact will different hyperparameter settings have on the weight distribution, for example, results like those in Figure 6?
7. Since M-VAE is a newly proposed dataset and lacks validation of its effectiveness, I think other widely used datasets such as UCF-Crime [a] and XD-Violence [b] should be added to further validate the performance of the model.

[a] Waqas Sultani, Chen Chen, and Mubarak Shah. Real-world anomaly detection in surveillance videos. In CVPR, 2018.

[b] Peng Wu, Jing Liu, Yujia Shi, Yujia Sun, Fangtao Shao, Zhaoyang Wu, and Zhiwei Yang. Not only look, but also listen: Learning multimodal violence detection under weak supervision. In ECCV, 2020.

8. The Non-LLM Models compared in the paper are not up-to-date enough. Meanwhile, the LLM-based models should also be compared on CUVA.
9. The p-value doesn't appear in Table 2, yet it is mentioned multiple times in the analysis. I think additional results should be added to the table to enhance its credibility.
10. Due to the complex architecture of the model, insufficient consideration has been given to the difficulties and costs of implementing the model in practical applications. Issues such as the computing resources required by the model, the cost of data labeling, and the difficulty of integrating with existing monitoring systems should be discussed in detail.
11. Past research VAD methods regarding the strategies of using imbalance optimization and local-global optimization should be added in the related work. Meanwhile, it should be illustrated what advantages Sherlock has compared to previous technologies.

**Reviewer Confidence:**

4: The reviewer is certain that the evaluation is correct and very familiar with the relevant literature

**Scope:**

3: The work is somewhat relevant to the Web and to the track, and is of narrow interest to a sub-community

---

### Official Review · Reviewer_iYzY · 2024-11-25

**Novelty:** 4
**Technical Quality:** 5

**Review:**

Quality:
+ The paper may not sufficiently address how well the Sherlock model scales to different video lengths, resolutions, or diverse datasets beyond the ones tested. This could be a concern regarding the model's robustness and generalizability.
+ The introduction of a Global-local Spatial-enhanced MoE module and a Spatial Imbalance Regulator might increase the complexity of the model. If not managed properly, this could lead to overfitting or longer training times, which could be a drawback in practical applications.

Clarity:
+ While the paper is structured well, the use of complex technical jargon or acronyms without sufficient explanation might hinder understanding for readers from different backgrounds or those new to the field.
+ The paper could be improved with more detailed visualizations or examples to help readers better grasp the concepts and results, especially regarding the spatial information processing.

Originality:
+ While the paper claims originality in its approach, some aspects might not be as groundbreaking as they appear, with similarities to existing models in the literature. The incremental innovation over prior work could be more clearly articulated.
+ The originality of the M-VAE task and dataset might be limited if similar tasks and datasets have been used in the literature, even if not as extensively. The paper could benefit from a more thorough comparison with existing datasets and tasks.

Significance:
+ The significance of the work could be diminished if the improvements over existing methods are marginal or if the practical benefits do not translate well into real-world applications due to computational or deployment constraints.
+ The long-term significance of the work might be limited if the approach is quickly superseded by new technologies or if the focus on spatial information becomes less relevant as video understanding research evolves.

**Questions:**

How does the Sherlock model perform on a diverse range of video datasets, particularly those with different cultural contexts or video qualities?

How do you ensure that the increased model complexity does not lead to overfitting, especially given the potential for a large number of parameters?

Will you be making the code and the dataset publicly available? If so, when can the community expect access?

How does the model handle edge cases or unusual events that were not well-represented in the training data?

**Reviewer Confidence:**

4: The reviewer is certain that the evaluation is correct and very familiar with the relevant literature

**Scope:**

4: The work is relevant to the Web and to the track, and is of broad interest to the community

---

### Official Review · Reviewer_3jCy · 2024-11-30

**Novelty:** 3
**Technical Quality:** 5

**Review:**

**Summary:**

This manuscript introduces a new video anomaly detection task which aims to extract semantic information of abnormal events from videos, including subjects, event types, objects, scenes, and locations. The proposed method Sherlock consists of two main components: (1) a global-local spatial enhance MoE for addressing the challenges on combining global and local spatial features, and (2) a spatial imbalance regulator for handling the imbalances in data distribution. In addition, the authors construct a dataset with semantic and spatial information of abnormal video records for the proposed task. The experimental results can confirm the effectiveness of Sherlock to some extent.


**Strong points:**

- S1: The proposed video anomaly detection task is both novel and practical, which could benefit real applications such as home security and online video content detection.

- S2: The constructed dataset for the problem will influence the development of this research direction, which is highly significant.


**Weak points:**

- W1: The proposed method resembles more of an engineering solution that integrates several existing techniques, lacking a certain level of novelty. The approach of modelling global and local information separately and then integrating them has been widely adopted in various fields of CV and NLP for several years. As for the integration method, most approaches now utilize learnable fusion techniques, which bear some similarity to the proposed spatial imbalance regulator.

- W2: The proposed model integrates a series of components, each designed to process different features. I am concerned that the inference latency of the model might be high, which could highly affect its practicality.

- W3: Some of the techniques used in the proposed model are empirically driven and lack theoretical support.

**Questions:**

- Q1: Section 4.4 provides a brief overview of the training and inference times. Could you add an experiment that offers a more detailed comparison of the average end-to-end inference time for each method on video inputs?

- Q2: Could you add theoretical analysis to support the loss used to balance the two types of spatial information?

- Q3: Could you clarify what the general global information is in line 124? This is the first time that the global information is mentioned, while it is not explained what it represents.

- Q4: There are 17 human joint nodes used in the spatial action tokens, mentioned in line 335. How are these 17 human joint nodes obtained, and how are they defined?

- Q5: What is the re-implementation for the baseline methods (cf. line 605)? Are the baseline methods fine-tuned with the instruction tuning dataset used in stage 2 of Sherlock?

- Q6: What is the variance?

**Reviewer Confidence:**

3: The reviewer is confident but not certain that the evaluation is correct

**Scope:**

3: The work is somewhat relevant to the Web and to the track, and is of narrow interest to a sub-community

---

### Official Review · Reviewer_mzEX · 2024-12-01

**Novelty:** 4
**Technical Quality:** 5

**Review:**

Advantages:
1. This paper proposes a multi-scenario video abnormal event extraction and localization (M-VAE) task and establishes a new dataset and benchmark for this task.
2. The paper clearly introduces the definition of the new task, the construction of the dataset and the details of the model.
3. This paper designs two corresponding modules for the benchmark model based on the two challenges of this task: GSM and SIR.
4. This paper has a positive impact on the field of information extraction and social security.

Disadvantages:
1. The author designed a GSM for the baseline model, which includes behavioral features, relationship features, background features and global features. Although the ablation experiment confirmed that any of these four features will lead to a decrease in the performance of the model. But I want to know, why are these four features effective for extracting abnormal events from videos? Or are there other more features that are helpful for extracting abnormal events?
2. Each expert in the GSM proposed by the author is basically based on one or two existing pre-trained models, which makes the entire model quite complicated. Is there a simpler and more effective way to extract these features? Or are these experts just guiding the knowledge in these pre-trained models?

**Questions:**

See the shortcomings of the review

**Reviewer Confidence:**

4: The reviewer is certain that the evaluation is correct and very familiar with the relevant literature

**Scope:**

4: The work is relevant to the Web and to the track, and is of broad interest to the community

---

### Official Review · Reviewer_yHFr · 2024-12-03

**Novelty:** 6
**Technical Quality:** 6

**Review:**

Authors propose the Multi-scene Video Abnormal Event Extraction and Localization (M-VAE) task. This involves localizing abnormal events in videos by determining timestamps and extracting structured event quadruples (subject, event type, object, and scene). This task emphasizes a structured approach to video anomaly detection, addressing gaps in existing methods that primarily classify individual frames as normal or abnormal. To tackle this task, the authors propose Sherlock, a Global-local Spatial-sensitive Large Language Model (LLM) that handles two challenges -- Global-local Spatial Modeling, and Global-local Spatial Balancing. Sherlock outperforms several state-of-the-art models in event extraction, localization, and anomaly classification metrics.

**Questions:**

1. How does Sherlock handle ambiguous and missing spatial information in videos?
2. What are the model’s limitations when applied to video domains not included in training?
3. How was the inter-annotator agreement ensured during dataset labeling, particularly for subjective categories like "event type"?
4. Can Sherlock effectively process videos with overlapping or simultaneous anomalies?
5. Is the model capable of real-time anomaly detection, and what are the computational requirements for such deployment?

**Reviewer Confidence:**

3: The reviewer is confident but not certain that the evaluation is correct

**Scope:**

3: The work is somewhat relevant to the Web and to the track, and is of narrow interest to a sub-community